# Nicotinamide Supplementation Mitigates Oxidative Injury of Bovine Intestinal Epithelial Cells through Autophagy Modulation

**DOI:** 10.3390/ani14101483

**Published:** 2024-05-16

**Authors:** Yihan Guo, Changdong Feng, Yiwei Zhang, Kewei Hu, Chong Wang, Xiaoshi Wei

**Affiliations:** 1College of Animal Science and Technology & College of Veterinary Medicine, Zhejiang A&F University, Hangzhou 311300, China; 2023608032011@stu.zafu.edu.cn (Y.G.); 2020608021008@stu.zafu.edu.cn (C.F.); zhangyiwei520@stu.zafu.edu.cn (Y.Z.); hukw123@stu.zafu.edu.cn (K.H.); 2Institute of Animal Health Products, Zhejiang Vegamax Biotechnology Co., Ltd., Anji 313300, China

**Keywords:** oxidative injury, intestinal oxidative homeostasis, intestinal epithelial cell, nicotinamide, autophagy, tight junction

## Abstract

**Simple Summary:**

The small intestine of ruminants is an important organ in the digestion and absorption of rumen indigestible nutrients while it is prone to oxidative stress, causing barrier damage, thus intestinal absorption and metabolic disorders. Nicotinamide was previously found to have antioxidant properties. The aims of this study were to investigate the effect of nicotinamide on intestinal oxidative damage and explore its potential mechanism. The study revealed that nicotinamide effectively reduced oxidative damage in bovine intestinal epithelial cells through autophagy. This protective effect was evidenced by enhanced antioxidant capacity and increased expression of tight junction proteins, suggesting nicotinamide’s role in improving cellular defense and intestinal barrier integrity.

**Abstract:**

The small intestine is important to the digestion and absorption of rumen undegradable nutrients, as well as the barrier functionality and immunological responses in ruminants. Oxidative stress induces a spectrum of pathophysiological symptoms and nutritional deficits, causing various gastrointestinal ailments. Previous studies have shown that nicotinamide (NAM) has antioxidant properties, but the potential mechanism has not been elucidated. The aim of this study was to explore the effects of NAM on hydrogen peroxide (H_2_O_2_)-induced oxidative injury in bovine intestinal epithelial cells (BIECs) and its potential mechanism. The results showed that NAM increased the cell viability and total antioxidant capacity (T-AOC) and decreased the release of lactate dehydrogenase (LDH) in BIECs challenged by H_2_O_2_. The NAM exhibited increased expression of catalase, superoxide dismutase 2, and tight junction proteins. The expression of autophagy-related proteins was increased in BIECs challenged by H_2_O_2_, and NAM significantly decreased the expression of autophagy-related proteins. When an autophagy-specific inhibitor was used, the oxidative injury in BIECs was not alleviated by NAM, and the T-AOC and the release of LDH were not affected. Collectively, these results indicated that NAM could alleviate oxidative injury in BIECs by enhancing antioxidant capacity and increasing the expression of tight junction proteins, and autophagy played a crucial role in the alleviation.

## 1. Introduction

For ruminants, the small intestine serves crucial roles in the digestion and absorption of ruminal undegradable starches and proteins [1] and maintains physical and biochemical barrier function and immune homeostasis [2,3]. It has been found that intestinal homeostasis in ruminants is maintained by the migration and proliferation of continuously self-renewing epithelial cells [4,5]. However, the small intestine is susceptible to oxidative stress, especially in the weaning period, and the accumulation of reactive oxygen species (ROS) could cause barrier dysfunction and intestinal damage [6]. Therefore, protecting the small intestine from oxidative stress is crucial.

Nicotinamide (NAM) is an amide form of vitamin B3. The NAM promoted the redox reaction of NAD + and increased the ratio of NAD + to NADH, thereby alleviating oxidative stress and avoiding apoptosis [7]. Previously, we have found that NAM could reduce the ROS level in periparturient dairy cows [8], human fibroblasts [9], and decrease hydrogen peroxide (H_2_O_2_) concentration in periparturient dairy goats [10]. However, the potential mechanism of NAM in alleviating oxidative stress has not been elucidated. Autophagy is an important intracellular lysosomal degradation pathway that protects intracellular homeostasis [11]. The accumulation of free radicals leads to oxidative damage and initiates the cellular autophagy process [12]. Yue et al. [13] found that impaired autophagy in mammary epithelial cells led to increased oxidative stress. Mitochondria are organelles capable of initiating and regulating autophagy [14]. We also found that the hepatic activities of mitochondrial complexes I and III were increased by NAM supplementation, indicating an improved mitochondrial respiratory chain status [10]. Therefore, we speculated that autophagy might be involved in the alleviation by NAM supplementation.

We hypothesized that NAM could attenuate intestinal oxidative injury in ruminants. To test the hypothesis, the bovine intestinal epithelial cells (BIECs) were used as experimental material, and H_2_O_2_ was used to induce oxidative stress. The effects of NAM on BIECs challenged by H_2_O_2_ and the potential protective mechanism were explored.

## 2. Materials and Methods

### 2.1. Materials

The cells used in this experiment were BIECs taken from jejunal segments, purchased from the Saizi (Shanghai) Bioengineering Co. (Shanghai, China) and stored in liquid nitrogen. The cells have been shown to be the bovine small intestinal epithelial cells and were used in previous studies [4]. Live animals were not used in this study. The NAM was purchased from the Sigma-Aldrich, St. Louis, MO, USA. Dulbecco’s modified Eagle’s medium nutrient mix (DMEM/F12) incomplete medium and phosphate buffer solution were obtained from the Shanghai Sangong Bioengineering Co. (Shanghai, China). Fetal bovine serum and 100× penicillin–streptomycin solution were from the Hangzhou Haotian Biotechnology Co. (Shanghai, China).

### 2.2. Cell Culture

The primary BIECs were isolated and cultured according to our previous report [1]. Briefly, the cells were cultured in DMEM/F12 medium (Sangon, Shanghai, China) supplemented with 10% fetal bovine serum (Sijiqing, Hangzhou, China) and 1% 100× penicillin-streptomycin solution (Sangon, Shanghai, China). The resuscitated cells were dispersed evenly and cultured in an incubator (Thermo Fisher, Waltham, MA, USA) at 37 °C with 5% CO_2_. The cells were passaged routinely with the cell confluence ranging from 85% and 95%, and the third passage adherent cells were used in the following experiments. In this study, every experiment was repeated 3 times, and within each date, 3 wells were performed in each treatment [1].

### 2.3. Effect of Nicotinamide on Bovine Intestinal Epithelial Cells Mediated by H_2_O_2_

This experiment aimed to investigate the effect of NAM on oxidative stress in BIECs and determine the optimal concentration of NAM for further experiments. The NAM (purity ≥ 99.5%, N0636-100G; Sigma, St Louis, MO, USA) was dissolved with the medium. First, the BIECs were incubated with NAM at 0, 1, 2, 4, and 8 mM, and the cell viability was analyzed. Hydrogen peroxide (H_2_O_2_) was widely used to induce oxidative injury [1,15]. Thus, the BIECs were treated with H_2_O_2_ at 0.5 mM in the absence or presence of NAM (0, 1, 2, 4, 8 mM) for 12 h. The cell viability, lactate dehydrogenase (LDH) concentration of cell supernatant, and intracellular total antioxidant capacity (T-AOC) were measured.

Subsequently, according to the results mentioned above, NAM at 2 and 4 mM were chosen for further analysis. Thus, the treatments were control (CON), oxidative stress (OS), OS + 2 mM NAM, and OS + 4 mM NAM. The relative gene and protein expressions of antioxidant enzymes (catalase, CAT, glutathione peroxidase 1, GPX1, manganese superoxide dismutase, SOD2), and tight junction proteins (Occludin, Claudin-2, and ZO-1), total autophagy, the protein expression of Beclin1, p62, and LC3-Ⅱ, and the mitochondrial potential membranes were measured. The specific measure methods are described below.

### 2.4. Cell Viability Assay

Cell viability was assessed using a Cell Counting Kit-8 assay (Boster, Wuhan, China) in accordance with the manufacturer’s protocol. Briefly, the cells were plated on 96-well plates with 100 μL per well and cultured for 24 h, then treated as described earlier for cell culture and treatment. Subsequently, 10% Cell Counting Kit-8 solution was added to each well of the plate and incubated for 2 h. Finally, the absorbance of each well was measured at 450 nm with a multimode microplate reader (BioTek, Winooski, VT, USA). All values were expressed as the proportion of control.

### 2.5. Determination of Lactate Dehydrogenase and Total Antioxidant Capacity

The LDH and T-AOC were determined using commercial kits in accordance with the kit protocols provided by the Nanjing Jiancheng Biotechnology Institute (Nanjing, China). Bicinchoninic acid protein assay reagent purchased from the Shanghai Biyuntian Biotechnology Co., Ltd. (Shanghai, China) was used to determine total cellular protein concentration. The absorbances were detected at 450 nm (LDH and T-AOC) and 562 nm (protein concentration) using a multimode microplate reader (BioTek, Winooski, VT, USA).

### 2.6. RNA Extraction and Real-Time Quantitative PCR

Total RNA was extracted from treated cells using a Total RNA Extraction Reagent (Takara, Beijing, China) according to the instructions provided by the manufacturer. RNA concentration and purity were determined by measuring the absorbance at 260 and 280 nm using a NanoDrop 2000 spectrophotometer (Thermo Fisher Science Inc.). Reverse transcription was then performed with the Prime SCRIPT RT kit (Takara). Quantitative PCR was performed with a CFX96™ Real-Time System (Bio-Rad, Hercules, CA, USA) using the SYBR PreMix Ex Taq II kit (Takara). The PCR cycle procedure comprised a 20 μL reaction system (containing 6.4 μL of enzyme-free water, 0.8 μL of forward primer, 0.8 μL of reverse primer, 2 μL of cDNA, and 10 μL of SYBR premix Ex Taq). Denaturation was first performed at 95 °C for 1 min, followed by 40 cycles of amplification at 95 °C for 30 s and 58 °C for 60 s. The results were normalized to the expression of β-actin and calculated using the 2^−ΔΔCT^ method. The primers for the genes are shown in Table 1.

### 2.7. Protein Extraction and Western Blotting

After the incubation, the supernatant was removed and the total proteins were extracted from the cells using RIPA Lysis buffer, protease inhibitor cocktail (100×), and phosphatase inhibitor cocktail (100×) (Beyotime, Shanghai, China), and the nuclear protein were extracted using the cytosolic and cytoplasmic protein extraction kit and phenylmethylsulphonyl fluoride (Beyotime, Shanghai, China). The protein concentration was determined using a bicinchoninic acid protein assay kit (Beyotime, Shanghai, China). Samples were loaded and separated on 10 % sodium dodecyl sulfate polyacrylamide gels and then transferred to the polyvinylidene difluoride (PVDF) membranes (Millikon, Burlington, MA, USA) via transmembrane buffer. The electrophoretic solution, transfer solution, and Tris-buffered saline with Tween 20 were prepared according to standard methods. The Western blotting instruments were purchased from Bio-Rad. The PVDF membranes were blocked in 5% albumin from bovine serum (BSA, dilution 1:20 with Tris-buffered saline with Tween 20) for 1 h at room temperature and incubated with primary antibodies at 4 °C with rocking. Primary antibodies against the following proteins were used: SOD2 (dilution 1:1000; Cell Signaling Technology, MA, USA), CAT (dilution 1:1000; Abclonal, Wuhan, China), ZO-1 (dilution 1:5000, Proteintech, Wuhan, China), Occludin (dilution 1:2000, Proteintech), Claudin-2 (dilution 1:1000, Proteintech), LC3 (dilution 1:1000, Abmart, Shanghai, China), p62 (dilution 1:1000, Cell Signaling Technology), Beclin1 (dilution 1:1000, Cell Signaling Technology), and β-actin (dilution 1:5000, Abclonal). After washing 3 times with washing buffer (Tris-buffered saline with Tween 20) for 10 min with rocking, the membranes were further incubated with the goat anti-rabbit immunoglobulin G horseradish peroxidase-conjugated secondary antibody (dilution 1:10,000, Abclonal) for 1 h at room temperature with rocking. The PVDF membranes were incubated with enhanced chemiluminescence substrate luminescent solution, and the blots were visualized with enhanced chemiluminescence (Beyotime, Shanghai, China) immunoblotting detection system. The images were obtained with a gel imaging system (Tanon, Shanghai, China). Finally, the relative quantity of protein bands was quantified by Image J software 1.53 (NIH, Bethesda, MD, USA) and normalized to β-actin protein in each sample.

### 2.8. Detection of Autophagy and Mitochondrial Membrane Potential

The cellular autophagy levels and mitochondrial membrane potentials were detected according to the kit instructions. The autophagy staining assay kit was purchased from the Shanghai Biyuntian Biotechnology Co., Ltd. (Shanghai, China). Briefly, the harvested cells were stained using monodansylcadaverine staining, incubated at 37 °C for 1 h away from light, and washed using assay buffer. The mitochondrial membrane potential assay kit was purchased from the Ebenezer Biotechnology Co. (Shanghai, China). The positive control was set up using carbonyl cyanide 3-chlorophenylhydrazone (CCCP) at 10 mM. The collected cells were stained using JC-1 staining and incubated at 37 °C for 40 min, and the images were captured using a fluorescence microscope (Carl Zeiss AG, Oberkochen, Baden-Württemberg, Germany).

### 2.9. Immunofluorescence

Briefly, the treated cells were fixed in 4% paraformaldehyde for 40 min, washed 2 times, and permeabilized with 0.3% TritonX-100 for 10 min at room temperature. Then cells were blocked with 5% closure solution for 2 h at room temperature, incubated with primary antibody (1:50 dilution) overnight at 4°C protected from light, and then washed. After that, the fluorescent secondary antibody (1:1000 dilution) was added and incubated at room temperature for 1 h, then stained with DAPI solution at room temperature for 10 min. Samples were sealed, and fluorescence images were captured using fluorescence microscopy [16].

### 2.10. Inhibition of Autophagy in Bovine Intestinal Epithelial Cells

Having established that autophagy occurring in BIECs can be mediated by H_2_O_2_, this study aimed to explore the role of autophagy in the alleviation of oxidative stress by NAM. Based on a prior study, chloroquine (CQ, Sigma-Aldrich) at 100 μM was used as an autophagy inhibitor to block cellular autophagy [17]. Cells were treated with/without CQ (100 μM), and/or H_2_O_2_ (0.5 mM), and/or NAM (2 and 4 mM) for 12 h. The LDH concentration of cell supernatant, intracellular T-AOC, and the protein expressions of LC3-Ⅱ and p62 were measured.

### 2.11. Statistical Analysis

The distribution of normality was evaluated by the Shapiro–Wilk test, and the homogeneity of variances was evaluated by the Levene test before the analysis. The data were analyzed using one-way ANOVA (SPSS 21.0), and Duncan’s multiple comparison tests were used to compare differences among groups if a significant treatment effect was observed. The results of analyses were input into GraphPad Prism 9.0 (GraphPad Software, Inc., San Diego, CA, USA) for graphical display. Data were expressed as mean ± standard error of the mean. The *p* ≤ 0.05 was considered to be statistically significant.

## 3. Results

### 3.1. Effect of Nicotinamide on Oxidative Stress Bovine Intestinal Epithelial Cells Challenged by H_2_O_2_

NAM at 1, 2, 4, and 8 mM all increased (*p* < 0.05) the cell viability of BIECs (Figure 1A). The H_2_O_2_ treatment decreased (*p* < 0.05) the cell viability of BIECs, and NAM at 2 mM increased (*p* < 0.05) it but not at 4 and 8 mM (Figure 1B). The amount of LDH released in the medium was greater in BIECs treated with H_2_O_2_ and decreased (*p* < 0.05) with NAM (1, 2, 4, and 8 mM) (Figure 1C). Similarly, the ability of T-AOC was decreased (*p* < 0.05) when challenged by H_2_O_2_, and NAM (1, 2, 4, and 8 mM) increased the T-AOC (Figure 1D, *p* < 0.05). Based on the results above, NAM at 2 and 4 mM was further used in the subsequent analyses.

### 3.2. Antioxidant Enzymes

To further investigate the effects of NAM on oxidative stress induced by H_2_O_2_, the expression of related genes and proteins was examined. Compared with the CON, the gene and protein expressions of CAT were significantly reduced (*p* < 0.05) by H_2_O_2_ treatment, and the NAM at 4 mM increased (*p* < 0.05) the gene and protein expressions of CAT to a similar level to CON (Table 2 and Figure 2A,B). The H_2_O_2_ had a tendency to decrease (*p* = 0.087) the gene expression of SOD2 (Table 2), and the protein expression of SOD2 was significantly decreased (Figure 2A,C, *p* < 0.05). There was no difference in the gene expression of GPX1 among groups.

### 3.3. Tight Junction Proteins

Compared with the CON, the gene expression of Claudin-2 was down-regulated (*p* < 0.05) with H_2_O_2_ treatment, and NAM at both 2 and 4 mM up-regulated (*p* < 0.05) it to a similar level as CON (Table 2). The results of the Western blot showed that the protein expressions of Occludin, Claudin-2, and ZO-1 were down-regulated (*p* < 0.05) by H_2_O_2_ (Figure 3). The NAM at both 2 and 4 mM up-regulated (*p* < 0.05) expression of Occludin and Claudin-2 to a similar level to CON, and the protein expression of ZO-1 was increased by NAM at 2 mM only (*p* < 0.05).

### 3.4. Effect of Nicotinamide on Autophagy of Bovine Intestinal Epithelial Cells Challenged by H_2_O_2_

Compared with the CON, the level of cellular total autophagy in the OS group was significantly increased (*p* < 0.05) and decreased (*p* < 0.05) with NAM at 4 mM the total autophagy (Figure 4A). The H_2_O_2_ treatment significantly increased (*p* < 0.05) the expression of proteins (Beclin1, p62, and LC3) compared with the CON, and the expression of these proteins was significantly decreased (*p* < 0.05) by NAM at 4 mM (Figure 4B). The immunofluorescence staining result of LC3 showed a significant increase (*p* < 0.05) in punctate immunosignals (representing autophagosomes) of BIECs in the OS group, and the immunosignals were significantly decreased (*p* < 0.05) by NAM at both 2 and 4 mM to a similar level as CON (Figure 4C).

Results on the change of mitochondrial membrane potential showed that the red/green ratio was lower (*p* < 0.05) in the positive control (CCCP) and OS groups compared with the CON, and it was increased (*p* < 0.05) by NAM at both 2 and 4 mM compared with the positive control and OS groups (Figure 5).

### 3.5. Effect of Autophagy Inhibitor and Nicotinamide on Bovine Intestinal Epithelial Cells Challenged by H_2_O_2_

The effect of autophagy inhibitors on autophagy was evaluated first. Without the autophagy-specific inhibitor (CQ), the effect of NAM treatment on protein expression of LC3 and p62 was consistent with the results shown in Figure 4B. When incubated with the CQ, the NAM at both 2 and 4 mM did not alter (*p* > 0.05) the protein expression of LC3 with/without H_2_O_2_ treatment (Figure 6A,B), and the protein expression of p62 was not changed (*p* > 0.05) by NAM at 2 mM while decreased by NAM at 4 mM with/without H_2_O_2_ treatment (Figure 6A,C).

Without the autophagy-specific inhibitor (CQ), the LDH concentration and T-AOC ability were consistent with the results shown in Figure 1. The LDH concentration was not altered (*p* > 0.05) by NAM at 2 or 4 mM with/without H_2_O_2_ treatment (Figure 6D), and so was the T-AOC ability (Figure 6E, *p* > 0.05).

## 4. Discussion

The ruminant intestine is a pivotal site for digestion and absorption of post-rumen nutrients. Extensive research has shown that oxidative stress can cause damage to the intestine and trigger apoptosis in the epithelial cells of ruminants [4,18,19]. Imbalances in redox levels within these cells lead to oxidative stress, but dietary supplementation with antioxidants has been found to alleviate intestinal inflammation and mucosal damage [20]. NAM, known for its ability to mitigate oxidative stress, has been previously studied in this context [21]. The findings of this study align with these established studies.

Our research demonstrated that NAM conferred protection against oxidative stress in BIECs, preserving cellular activity under stress conditions. The decrease in the cell survival rate after H_2_O_2_ treatment and the increase in the cell survival rate after NAM treatment proved its protective effect. NAM supplementation resulted in a decrease in intracellular LDH and increases in the T-AOC, CAT, and SOD2, suggesting that NAM had a positive effect on antioxidant activity. Wei et al. [1] found that H_2_O_2_-induced oxidative stress in BIECs resulted in an increase in the concentration of LDH and decreases in T-AOC and the activity of antioxidant enzymes. Additionally, previous reports indicated that NAM supplementation in periparturient cows could improve serum oxidative status and glutathione metabolism, alleviating oxidative stress in cows [8]. These results further demonstrated that NAM could alleviate oxidative stress by regulating intracellular antioxidant enzymes, thus protecting the cells.

Intestinal integrity, crucial for proper intestinal function, relies on tight junction proteins, which play a major role in maintaining the intestinal barrier. Occludin, Claudin-2, and ZO-1 are among the key tight junction proteins, and their expression levels serve as important indicators of intestinal barrier function [22,23]. When the intestine is exposed to oxidative stress, accumulated free radicals cause damage to tight junction proteins, thereby disrupting the intestinal barrier [24]. In the present study, NAM significantly inhibited the H_2_O_2_-induced reduction in the expression levels of Occludin, Claudin-2, and ZO-1 in BIECs. NAM, the amide form of niacin, has been shown to increase the relative gene expression of Occludin and Claudin-1 in the small intestine when supplemented in weaned piglets [25]. Consistent with our study, Li et al. [26] found that the addition of nicotinamide adenine dinucleotide increased the expression level of intestinal tight junction proteins and alleviated intestinal barrier damage in mice.

Autophagy, a cellular process involving the degradation and recycling of cellular components, is marked by proteins such as LC3, Beclin1, and p62 [27]. In the present study, we observed a significant upregulation of LC3 and Beclin1 expressions in BIECs induced by H_2_O_2_, indicating that oxidative stress likely triggers cellular autophagy. It has been demonstrated that LC3-II expression was upregulated in cells induced by oxidative stress [27]. Zhao et al. [28] found that H_2_O_2_ induced a significant increase in LC3-II levels in Naked mole-rats skin fibroblasts and hepatic stellate cells. The Beclin1 expression was also up-regulated in cells treated with H_2_O_2_ [29]. In addition, autophagy plays a role in clearing ubiquitinated proteins, with p62 being involved in this process [30,31]. Our results found that oxidative stress led to increased levels of p62 expression, speculated to be caused by autophagic flow blockage and p62 accumulation. These results confirmed our hypothesis that oxidative stress induces cellular autophagy. Autophagy relieves oxidative stress, but excessive autophagy causes cellular damage. In this study, we found that the expression of LC3-II was down-regulated by 2 mM NAM treatment, and the expression of LC3-II, Beclin1, and p62 was down-regulated by 4 mM NAM treatment, which suggested that NAM attenuated oxidative stress-induced autophagy. Based on these observations, it is plausible to propose that NAM reduced oxidative stress-related cellular damage by modulating autophagy levels.

The decline in mitochondrial membrane potential serves as an early indication of mitochondrial membrane damage and apoptosis onset, as an imbalance in mitochondrial membrane potential precedes apoptosis [32]. In our study, the cellular mitochondrial membrane potential was significantly decreased by H_2_O_2_ treatment, indicating an alteration in osmotic pressure and a reduction in the potential difference between the inner and outer mitochondrial membranes. Previous research has shown that acute exposure of mouse cardiomyocytes to H_2_O_2_ induced slow fragmentation of mitochondria, resulting in mitochondrial damage [33]. Similarly, in porcine intestinal epithelial cells, H_2_O_2_ decreased mitochondrial membrane potential, which was restored upon alleviation of oxidative stress [34]. In our experiments, we observed that NAM effectively prevented the decrease in mitochondrial membrane potential induced by H_2_O_2_. This suggested that NAM had the potential to mitigate oxidative stress-induced mitochondrial membrane damage, thereby offering a protective effect against cellular stress.

In this study, we observed that NAM protected BIECs from H₂O₂-induced oxidative barrier and mitochondrial damage. However, the detailed molecular mechanism underlying these effects required exploration. Oxidative stress is known to trigger autophagy, which can mitigate cellular damage. Nonetheless, excessive autophagy under oxidative stress conditions can exacerbate cellular injury. We found that the protein expression levels of LC3-II and p62 were increased after the use of an autophagy inhibitor. The entry of autophagy markers into the autophagosome is a clear indicator of autophagy, and inhibition of autophagy by CQ resulted in the accumulation of autophagy markers and inability to enter the autophagosome, demonstrating that the autophagy pathway is blocked [35]. It has been demonstrated that CQ treatment induced the accumulation of autophagy labeling sites [36]. Bik et al. [37] found that the higher the concentration of CQ treatment, the more LC3-positive structures in cells. In adipocytes, the protein expression levels of LC3-II and p62 were increased in a time-dependent manner starting 3 h after CQ treatment [38]. After blocking the autophagy pathway, we found that the addition of NAM did not alter the protein expressions of LC3-II and p62, nor did it change the intracellular LDH concentration and T-AOC level. These data suggested that the protective effect of NAM against H_2_O_2_-induced oxidative injury in BIECs might be involved in the autophagy pathway.

## 5. Conclusions

The NAM could attenuate oxidative injury in BIECs by enhancing antioxidant capacity and increasing the expression of tight junction proteins. In addition, the NAM supplementation modulated the autophagy pathway, which played a key role in the mitigation of oxidative injury of BIECs. Our study contributed valuable data to understanding the regulatory mechanism of NAM in alleviating oxidative stress. Nevertheless, the absence of animal evidence could be considered a limitation of this study, and future research should include such trials to further validate these findings.

## Figures and Tables

**Figure 1 animals-14-01483-f001:**
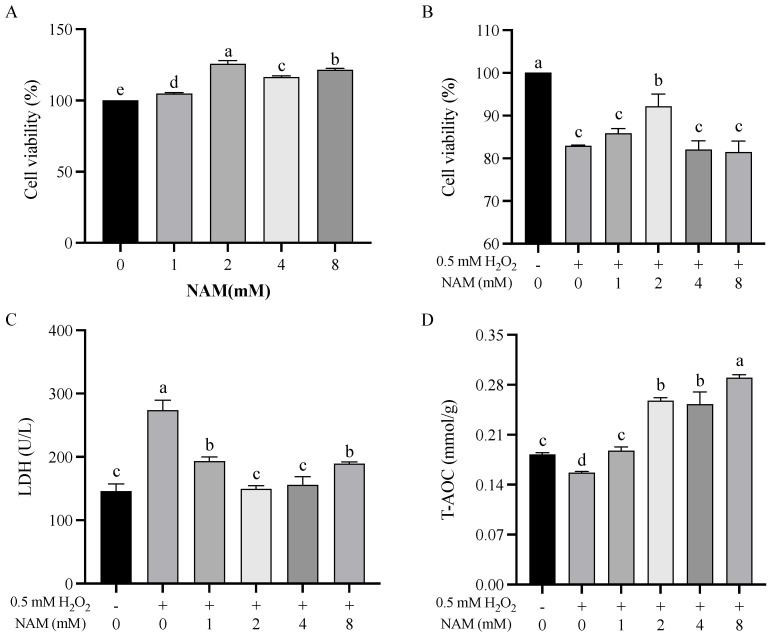
Study on the optimal concentration of nicotinamide (NAM). The bovine intestinal epithelial cells (BIECs) were treated with/or without H_2_O_2_ (0.5 mM) and/or NAM (0, 1, 2, 4, or 8 mM) for 12 h. (**A**) Cell viability of BIECs with NAM treatment. (**B**) Cell viability of BIECs with H_2_O_2_ and NAM treatment. (**C**) Lactate dehydrogenase (LDH) release in BIECs with H_2_O_2_ and NAM treatment. (**D**) Total antioxidant capacity (T−AOC) of BIECs with H_2_O_2_ and NAM treatment. Values are means and error bars represent standard error. Mean values with different letters are significantly different (*p* ≤ 0.05).

**Figure 2 animals-14-01483-f002:**
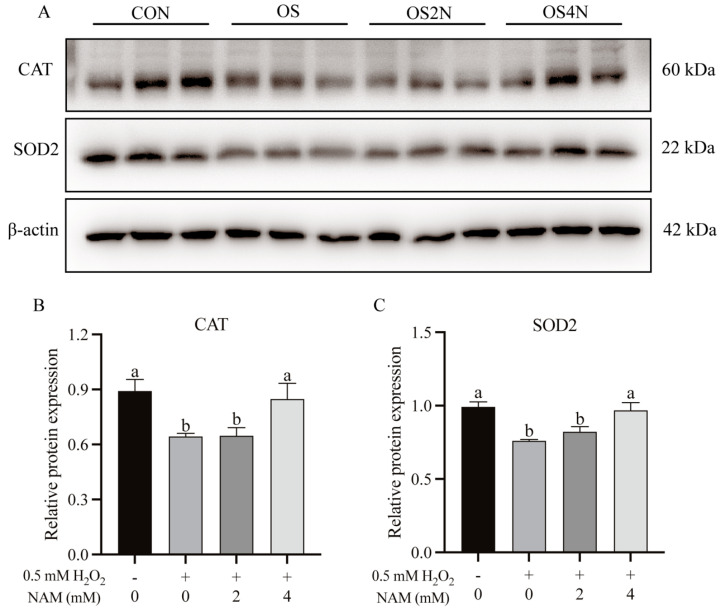
Effect of nicotinamide (NAM) on the relative protein expression of antioxidant enzymes in bovine intestinal epithelial cells (BIECs) mediated by hydrogen peroxide (H_2_O_2_). The BIECs were treated with H_2_O_2_ (0.5 mM) and/or NAM (2 and 4 mM) for 12 h. CON = control group, cells were treated without H_2_O_2_ nor NAM; OS, OS2N, OS4N = cells were treated with H_2_O_2_ (0.5 mM) and NAM (0, 2, and 4 mM) for 12 h, respectively. (**A**–**C**) Relative protein expression of catalase (CAT) and superoxide dismutase 2 (SOD2). Values are means and error bars represent standard error. Mean values with different letters are significantly different (*p* ≤ 0.05).

**Figure 3 animals-14-01483-f003:**
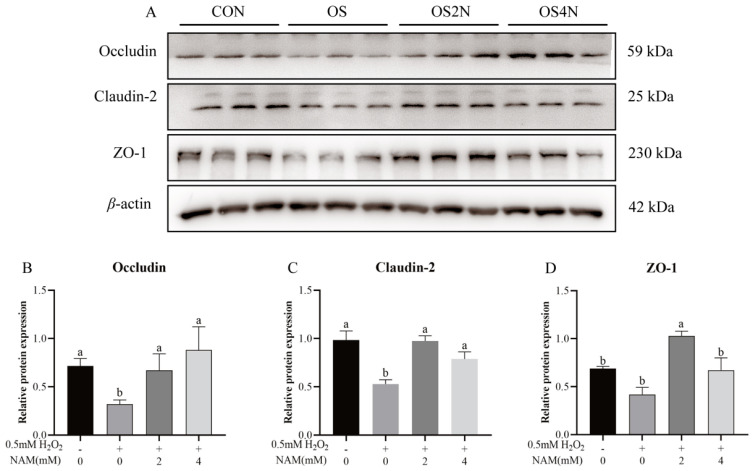
Effect of nicotinamide (NAM) on the relative protein expression of tight junction protein in bovine intestinal epithelial cells (BIECs) mediated by hydrogen peroxide (H_2_O_2_). The BIECs were treated with H_2_O_2_ (0.5 mM) and/or NAM (2 and 4 mM) for 12 h. CON = control group, cells were treated without H_2_O_2_ nor NAM; OS, OS2N, OS4N = cells were treated with H_2_O_2_ (0.5 mM) and NAM (0, 2, and 4 mM) for 12 h, respectively. (**A**–**D**) Relative protein expressions of Occludin, Claudin−2, and ZO−1. Values are means and error bars represent standard error. Mean values with different letters are significantly different (*p* ≤ 0.05).

**Figure 4 animals-14-01483-f004:**
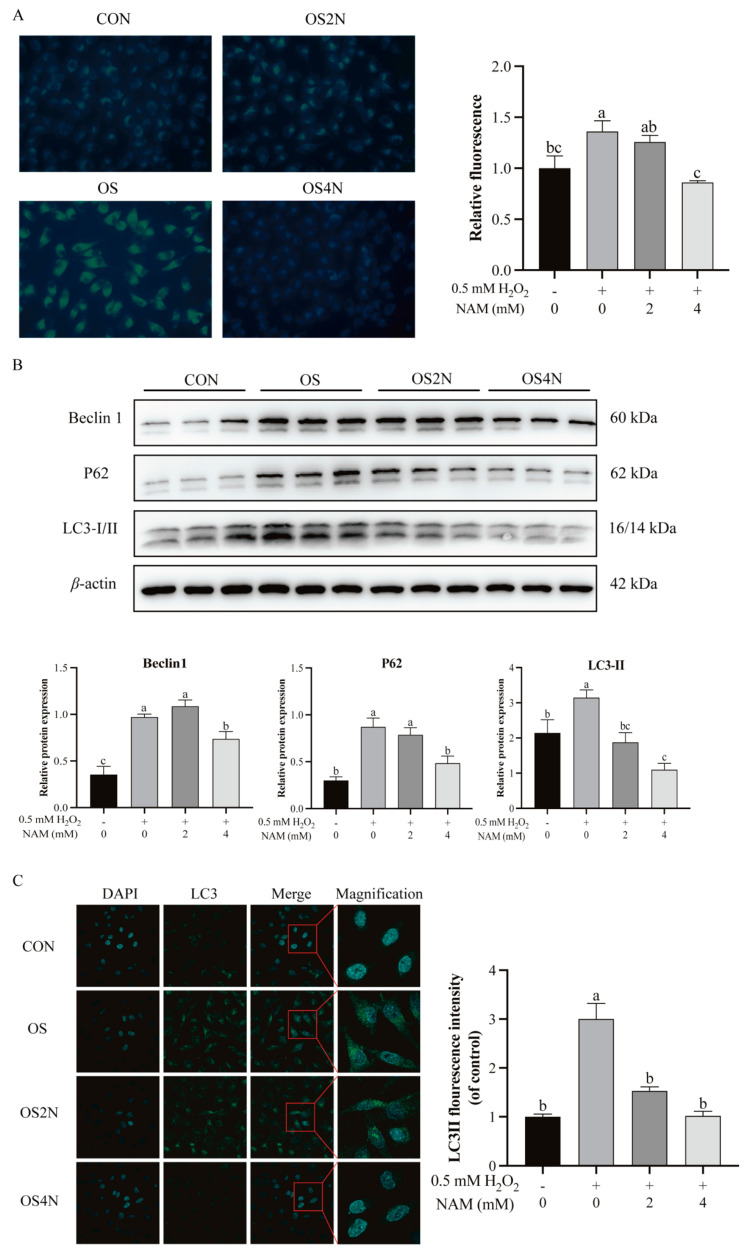
Effect of nicotinamide (NAM) on the autophagy in bovine intestinal epithelial cells (BIECs) mediated by hydrogen peroxide (H_2_O_2_). The BIECs were treated with H_2_O_2_ (0.5 mM) and/or NAM (2 and 4 mM) for 12 h. CON = control group, cells were treated without H_2_O_2_ nor NAM; OS, OS2N, OS4N = cells were treated with H_2_O_2_ (0.5 mM) and NAM (0, 2, and 4 mM) for 12 h, respectively. (**A**) Fluorescent staining levels and difference analysis of autophagy. (**B**) Relative protein expression of Beclin1, p62, and LC3−Ⅱ. (**C**) Punctate immunosignals and difference analysis of fluorescence quantization of LC3 protein. Values are means and error bars represent standard error. Mean values with different letters are significantly different (*p* ≤ 0.05).

**Figure 5 animals-14-01483-f005:**
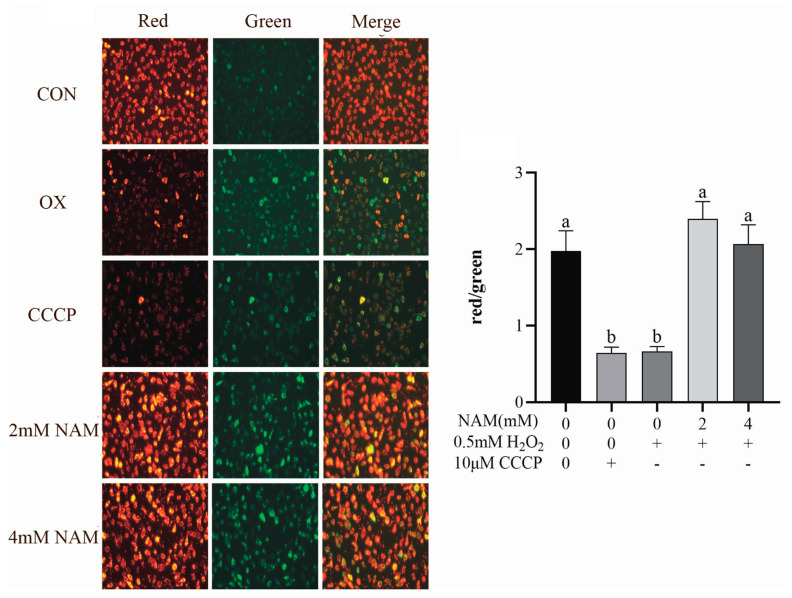
Effect of nicotinamide (NAM) on mitochondrial membranes potential in bovine intestinal epithelial cells (BIECs) mediated by hydrogen peroxide (H_2_O_2_). CCCP: the positive control. The red and green fluorescence staining diagram of mitochondrial membrane potential. Values are means and error bars represent standard error. Mean values with different letters are significantly different (*p* ≤ 0.05).

**Figure 6 animals-14-01483-f006:**
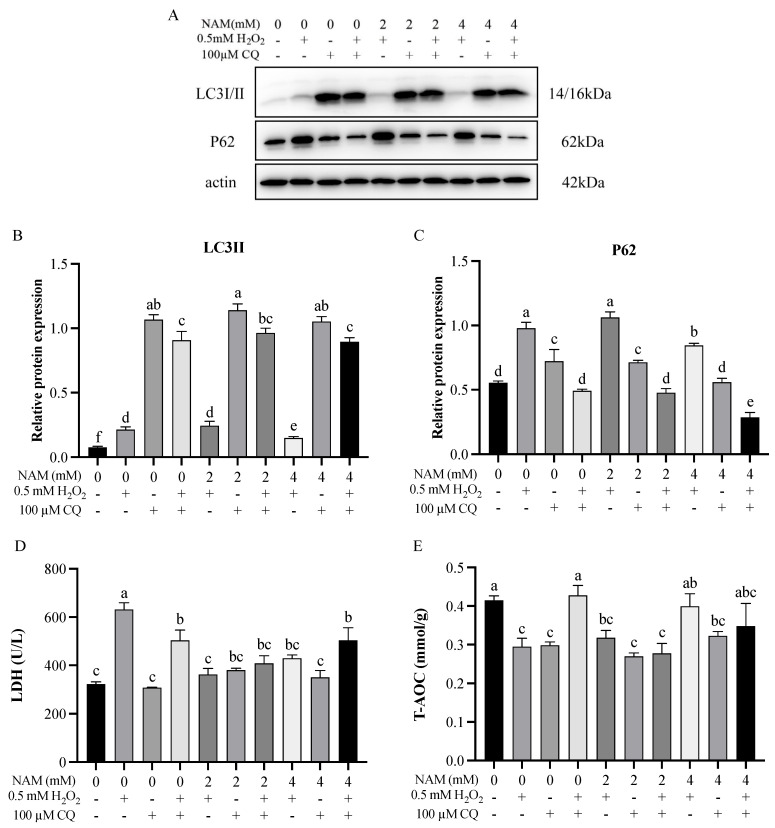
Nicotinamide (NAM) enhanced antioxidative capacity through autophagy in bovine intestinal epithelial cells (BIECs) mediated by hydrogen peroxide (H_2_O_2_). Cells were treated with/without chloroquine (CQ, 100 μM), and/or H_2_O_2_ (0.5 mM), and/or NAM (2 and 4 mM) for 12 h. (**A**–**C**) Relative protein expressions of LC3-Ⅱ and p62. (**D**,**E**) The lactate dehydrogenase (LDH) concentration and the total antioxidant capacity (T−AOC). Values are means and error bars represent standard error. Mean values with different letters are significantly different (*p* ≤ 0.05).

**Table 1 animals-14-01483-t001:** The primer sequences used in this study.

Gene ^1^	Primer Sequences (5′-3′)
*β*-actin	F: CGGGAAATCGTCCGTGAC R: CCGTGTTGGCGTAGAGGT
CAT	F: AGATACTCCAAGGCGAAGGTG R: AAAGCCACGAGGGTCACGAAC
GPX1	F: AGTGCGAGGTGARATGGCGAGARA R: TGGGCARAARATCCCTGGAGAGCA
SOD2	F: CGTCGCCGAGGAGAAGTA R: CCAGCAGGGGGATAAGA
Claudin-2	F: CCAGGCCATGATGGTGACAT R: GAAGAAGACTCCGCCCACAA
Occludin	F: ACGCAGGAAGTGCCTTTGGTAGC R: GCAGCCATGGCCAGCAGGAA
ZO-1	F: GAAAGATGTTTATCGTCGCATCGT R: ATTCCTTCTCATATTCAAAATGGGTTCTGA

^1^ *β*-actin: actin gamma 1; CAT: catalase; GPX1: glutathione peroxidase 1; SOD2: superoxide dismutase 2.

**Table 2 animals-14-01483-t002:** Effects of nicotinamide (NAM) on relative mRNA expression of antioxidant enzymes and tight junction proteins.

Items ^1^	Treatments ^2^				SEM	*p*-Value
	CON	OS	OS2N	OS4N		
Antioxidant enzymes						
CAT	1.01 ^a^	0.72 ^b^	0.87 ^ab^	1.09 ^a^	0.10	0.023
GPX1	1.00	1.20	1.22	1.11	0.14	0.436
SOD2	1.01	0.72	1.16	0.97	0.17	0.087
Tight junction proteins						
Occludin	1.00	0.76	0.93	0.94	0.05	0.258
Claudin-2	1.00 ^a^	0.62 ^b^	1.12 ^a^	0.96 ^a^	0.08	0.015
ZO-1	1.00	1.07	0.83	0.99	0.05	0.119

^1^ CAT: catalase; GPX1: glutathione peroxidase 1; SOD2: superoxide dismutase 2. ^2^ CON = control group, cells were treated without H_2_O_2_ nor NAM; OS, OS2N, OS4N = cells were treated with H_2_O_2_ (0.5 mM) and NAM (0, 2, and 4 mM) for 12 h, respectively. Data represent the mean ± SEM. ^a,b^ Mean values in columns without a common letter are significantly different (*p* < 0.05).

## Data Availability

The supporting data of this study are available within the article.

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
