# Peer review of "Nicotinamide Supplementation Mitigates Oxidative Injury of Bovine Intestinal Epithelial Cells through Autophagy Modulation"

_animals, 2024, doi:10.3390/ani14101483_

Round 1

Reviewer 1 Report

Comments and Suggestions for Authors

Well done on a nice manuscript. It is good to see work supporting the positive effects of Niacinamide.

Title Suggestion; Nicotinamide supplementation alleviates oxidative injury in bovine intestinal epithelial cells and the autophagic process plays a crucial role in the alleviation.

L12        undegradable?  while -> and   comma after damage

L18        The small

L20         delete the

L21         while -> but

L23         H2O2

L25         delete comma

L26         H2O2

L28         And -> Additionally,  delete the first the

L34         Injury

L38         ruminants

L39        it also

L40         barriers,

L42         delete target

L43         the weaning

L44         the small

L48        Previously, we

L51         are few -> is little

L62         maintain

L65         injury

L72         delete And

L89         delete was

L92         replace with ,

L94         injury

L97         delete the    to-> for

L101      delete the

L106      protocol of the manufacturer -> manufacturer’s protocol.

L109      , culturing for -> and incubated for  hole -> well

L114      the Nanjing  (and same for all other companies unless in brackets)

L121      a Total

L123      by -> with a or using a

L129      Amplification -> Denaturation

L130      cycles of amplification at

L144      electrophoretically

L146      to the laboratory formulas. -> standard methods

L147      purchased from

L149      was rocking or agitation used? (and for other incubations

L148      what was the concentration of BSA and what was the dilution buffer?

L161      by the -> with a

L165      potentials

L166      instructions of the commercial kits - kit instructions.

L173      a fluorescence microscope (manufacturer)

L178      then washed

L182      microscopy

L184      can be mediated

L186      delete the

L187      an autophagy

L194      differences

L200      delete first The

L201      The H2O2 treatment decreased the cell viability of BIECs, and NAM at 2 mM increased it (Figure

L202      P < 0.05) but not at 4 and 8 mM.

L203      H2O2, and decreased with NAM (1, 2, 4, and 8 mM) (Figure

L204      when challenged

L205      delete first the  T-AOC (Figure 1C).

L232      expression

L240      delete the   to -> as

L241      protein expressions or expression  (optional)

L243      protein expressions or expression  (optional)

L248      H202

L249      H202

L250      H202     H202

L253      Inconsistency throughout between spacings in (P≤0.05) and (P < 0.05)

L256      < 0.05), and decreased with  NAM at 4 mM the total 256 autophagy (P < 0.05)

L262      to simcompared to a similar level as CON

L301      per -> post

L348      produce -> exhibit

Figure appears a bit fuzzy and can be sharpened up using methods in the attachment.

Comments on the Quality of English Language

There are a lot of editorial corrections suggested in the non-discussion parts of the manuscript and a way of improving the sharpness of the figures is offered.

Reviewer 2 Report

Comments and Suggestions for Authors

Comments and Suggestions for Authors

After reviewing the manuscript entitled “The autophagy plays vital roles in alleviating oxidative injure of bovine intestinal epithelial cells by nicotinamide supplementation”, the following suggestions were made it. The manuscript is very interesting and provides novel information on the effects of nicotinamide on autophagy and oxidative damage of intestinal epithelial cells. The study uses scientifically valid methods; however, several details must be corrected before the manuscript can be considered for publication in this prestigious journal. Specific comments are listed below.

Simple Summary

Line 13: Please specify what type of disorders.

Line 13, 14, and 15: Abbreviations should not be used in the simple summary. Please replace the abbreviations “NAM” with the full name.

Abstract

Line 20: Please specify what type of symptoms and nutritional consequences.

Line 20: Delete “Our”.

Line 23: Please change “(H2O2)” to “(H2O2)”.

Line 24: Delete “The results of this study”. This phrase is obvious and should be avoided in the abstract.

Lines 24-30: The results shown in the abstract should be rewritten since they are very ambiguous in their current form. In these lines, the response variables (or at least the most important ones) that increased or decreased should be specified. Likewise, the significance level at which the changes were detected or at which no changes were detected must be specified.

Keywords: Oxidative injure; bovine intestinal epithelial cell; nicotinamide; autophagy. These words used as keywords are the same as those previously used in the title of the manuscript. Keywords should be different from those in the title (but related to the topic) to broaden the reach of academic search engines in case the manuscript is later published.

Introduction

Lines 41-43: Please add at least one sentence explaining how oxidative stress affects the migration and proliferation of continuously self-renewing epithelial cells.

Line 45: The authors should mention the main strategies that have been used to mitigate oxidative stress and prevent its effects on intestinal damage.

Line 49: The abbreviation “ROS” has not been previously specified.

Line 64: Please delete “[1,4,15]”.

Material and methods

Lines 75-86: Authors must justify why they used these cultivation methods, and this justification must be supported by relevant scientific references.

Line 99: The authors should clearly indicate which methods were used to measure changes in gene expression since several methods are currently available. After specifying this information, it must also be justified with relevant references explaining why those methods were used.

Lines 175-182: Authors must justify why they used these immunofluorescence methods, and this justification must be supported by relevant scientific references.

Lines 192-197: The authors must indicate which statistical tests they used to evaluate the data's normality and homogeneity of variance. This is very important to know if the correct tests were used. Also, could you explain why you used Duncan's test to compare the differences between treatments?

Results

Line 199: Please do not use abbreviations in the subheadings of any section.

Lines 201-288: The entire wording of the results section needs to be improved. The significance value obtained must be specified immediately after mentioning the direction of the effect. For example, the sentence “The H2O2 treatment decreased the cell viability, and NAM at 2 mM increased that of 201 BIECs (Figure 1B, P < 0.05)” should be changed to “The H2O2 treatment decreased (p < 0.05) the cell viability, and NAM at 2 mM increased (p < 0.05) that of 201 BIECs (Figure 1B)”.

Lines 201-288: The authors must modify the format used to indicate significance levels throughout the results section. The instructions for authors indicate that the p should be lowercase and italics. However, the authors used capital P.

Figure 1 A: Please change “Cell viability(%)” to “Cell viability (%)”.

Lines 248-250:  Please change “H2O2” to “H2O2”.

Lines 296-299: The paragraph on lines 296-299 should be changed immediately after paragraph 284-288.

Discussion

Lines 307-309: These lines must be removed. In the discussion section, the results found should be explained and contrasted with those obtained from similar studies. However, lines 307-309 only mention what was evaluated in the present study. That information was previously added to the end of the introduction section.

Line 318: Please delete “And”.

Lines 318-321: This sentence needs to be clarified; the authors should modify it and clearly indicate who found results consistent with those obtained in the present study. Likewise, the reference should be added immediately after mentioning the study with which the results are being contrasted and should not be placed until the end of the sentence (as it currently stands).

Line 322: It should be specified in what type of tissue from dairy cows the authors of the study reviewed and cited found an increase in antioxidant activity. This specification is important since intestinal cells were used in the present study.

Line 336: What antioxidants were used by Li et al.? Were they NAM? Otherwise, the authors should change the reference to another study where the same antioxidant (NAM) as in the current study was used.

Conclusion

Line 403: Please delete “Our data suggested that”.

Comments on the Quality of English Language

The manuscript needs a moderate grammar check as there are several details and some sentences are not clear.

Reviewer 3 Report

Comments and Suggestions for Authors

General Comment:

The manuscript titled "The Autophagy Plays Vital Roles in Alleviating Oxidative Injury of Bovine Intestinal Epithelial Cells by Nicotinamide Supplementation" submitted to Animals Journal explores the impact of Nicotinamide on oxidative stress induced by H2O2 in intestinal epithelial cells. I have several concerns regarding the manuscript which need to be addressed before I can recommend this article for publication.

Specific Comments:

1.        The clarity of language in the manuscript is inadequate, which significantly hinders comprehension. The title itself lacks clarity and impact, suggesting the need for a thorough language review throughout the manuscript.

2.        Correct the statement “For ruminant, the small intestine is an important site for the digestion and absorption of non-degradable starches and proteins from the rumen [1], also plays a vital role in physical and biochemical barrier and immune homeostasis [2,3].

3.        Please revise the sentence for clarity and grammatical accuracy: "For ruminants, the small intestine is an important site for the digestion and absorption of non-degradable starches and proteins from the rumen [1], and it also plays a vital role in physical and biochemical barrier functions and immune homeostasis [2,3]."

4.        Revised the sentence for clarity “Line 69-70, cow small intestinal epithelial cells (taken from jejunal segments, Bovine intestinal epithelial cells, BIECs),

5.        Could you provide details on how the bovine intestinal epithelial cells (BIECs) were stored and transported?

6.        The exclusion of Nrf2 and MDA as markers in the study is not justified. Please explain why these were not considered.

7.        On line 186, a reference to previous work is mentioned but not cited. Please provide the appropriate citation.

8.        Replace "oxidative injure" with "oxidative injury" throughout the manuscript 

9.        The manuscript focuses exclusively on oxidative stress. Please explain the rationale behind not assessing inflammatory responses alongside oxidative stress.

10.    The statement on the effects of Nicotinamide is unclear. Do you mean to describe "Effect of Nicotinamide on H2O2-mediated oxidative stress in BIECs?"

11.    Discussion and Conclusion: The conclusion and discussion sections require a comprehensive re-drafting to better reflect the findings and logically conclude the study.

12.    The manuscript must discuss the limitations of the study explicitly in the conclusion section.

Comments on the Quality of English Language

Extensive editing of English language required

Round 2

Reviewer 2 Report

Comments and Suggestions for Authors

The authors have responded satisfactorily to each of my suggestions. Therefore, I recommend acceptance of the manuscript.

Comments on the Quality of English Language

The quality of English is good

Author Response

Dear reviewer,

Thanks for your suggestion.  We have revised the English language throughout the manuscript. Hoping it is better now.

Reviewer 3 Report

Comments and Suggestions for Authors

The title, abstract, discussion, and conclusion sections are notably weak particularly from a scientific language perspective, and still require significant revision to clearly convey the implications and the scope of the research.

Comments on the Quality of English Language

Extensive editing of English language required

Author Response

Dear reviewer,

Thanks for your suggestion. We have revised the English language throughout the manuscript. Hoping it could be better and get your approval.

Round 3

Reviewer 3 Report

Comments and Suggestions for Authors

Title is not ok.

Suggested title: Nicotinamide Supplementation Mitigates Oxidative Injury of Bovine Intestinal Epithelial Cells through Autophagy Modulation

The last three lines of summary (16-19) and abstract (31-34) are duplicated.

Discussion and conclusion still need revision

Line 390-391 replace the sentence with "In addition, the NAM supplementation modulated the autophagy pathway, which played a key role in the mitigation of oxidative injury of BIECs".

Comments on the Quality of English Language

Moderate editing of English language required

Author Response

Dear reviewer, 

Thanks for your suggestion. We have revised the manuscript as you suggested. Please see the revised vision. Hoping it could get your approval.